# The Anatase-to-Rutile Phase Transition in Highly Oriented Nanoparticles Array of Titania with Photocatalytic Response Changes

**DOI:** 10.3390/nano12244418

**Published:** 2022-12-11

**Authors:** Olga Boytsova, Irina Zhukova, Artem Tatarenko, Tatiana Shatalova, Artemii Beiltiukov, Andrei Eliseev, Alexey Sadovnikov

**Affiliations:** 1Department of Materials Science, Lomonosov Moscow State University, Building 73, Leninskie Gory 1, Moscow 119991, Russia; 2Department of Chemistry, Lomonosov Moscow State University, Building 3, Leninskie Gory 1, Moscow 119991, Russia; 3Udmurt Federal Research Center of UB RAS, T. Baramzina Str. 34, Izhevsk 426067, Russia; 4Kurnakov Institute of General and Inorganic Chemistry RAS, Leninskii Prosp. 31, Moscow 119071, Russia; 5Topchiev Institute of Petrochemical Synthesis, Russian Academy of Sciences, Leninskii Prosp. 29, Moscow 119991, Russia

**Keywords:** nanophotocatalyst, mesocrystal, anatase, rutile, phase transition

## Abstract

An array of highly oriented anatase nanoparticles was successfully prepared from NH_4_TiOF_3_ with the assistance of polyetheleneglycol-400 at 450 °C. The study showed the stability of obtained layered TiO_2_-anatase close to 1200 °C. This research confirmed for the first time that the transition of mesocrystalline anatase to the rutile phase occurs between 1000 °C and 1200 °C, which is more than 400 °C higher than the transition of bulk TiO_2_ due to the used precursor. A small quantity of K-phase nanowhiskers, which issued after 800 °C in the composite based on TiO_2_, stimulated a fourfold increase in photocatalytic performance. This study offers a new approach to the construction and preparation of effective nanocrystalline photocatalyst.

## 1. Introduction

Titania nanoparticles are widely used in everyday products requiring photocatalytic and ion-exchange performance because of their specific surface growth and reactivity. It was found that in spite of the fact that myriad studies on TiO_2_-based nanomaterials have been made in recent decades, many scientists are interested in improving them further. Hence, the manipulation of phase stability and formation are essential for both researchers and industrial organizations. At room temperature and atmospheric pressure, the thermodynamically stable modification of TiO_2_ is rutile. Anatase and brookite can be kinetically stabilized. Single-crystal anatase irreversibly transforms into rutile at close to 600 °C in air; however, data are also available on transition temperatures in the range of 400–1200 °C due to differences in methods for determining transition temperatures, the presence of impurities, and variations in crystallite sizes. Thus, anatase mesocrystals are described in [1], which retain high stability up to 800–900 °C. The influence of impurities has also been repeatedly studied [2]. Brookite is more stable than anatase; the brookite–rutile phase transition occurs at 800 °C [2,3]. Anatase and rutile mesocrystals can exhibit higher photocatalytic activity due to their highly oriented ordered structure. In this case, it is important to have a structure with a significant proportion of (001) and (101) faces [4].

The literature describes several methods for obtaining titanium dioxide mesocrystals [5,6,7]. One of them is the annealing of NH_4_TiOF_3_ mesocrystals obtained by hydrolysis of (NH_4_)_2_TiF_6_ [5].

To stabilize the growth of the 001 or 101 faces, it is necessary to slow down the process of hydrolysis in other directions. To do this, non-ionic polymers, such as polyethylene glycol (PEG), are introduced into the reaction mixture. Then, only partial hydrolysis of TiF_6_^2−^ ions proceeds with the formation of an intermediate compound NH_4_TiOF_3_, which is stable and precipitates. Further, due to the similarity of the crystal structures of NH_4_TiOF_3_ and TiO_2_, as a result of heat treatment, TiO_2_ mesocrystals can be obtained [8].

Titanium dioxide highly oriented nanoparticles arrays can also be synthesized by simple solution methods. For example, hollow spheres from rutile nanorods can be synthesized by a hydrothermal reaction using TiCl_4_, *N*,*N*’-dicyclohexylcarbodiimide (DCC), and L-serine as biological additives [9]. Another way is to obtain anatase mesocrystals using solvothermal synthesis from (C_4_H_9_O)_4_Ti and acetic acid. The reaction was carried out for 24 h at 200 °C followed by annealing at 400 °C to remove organic residues [1]. Such highly porous mesocrystals can be produced in large quantities, and due to the high nanocrystallinity of the resulting material, the anatase phase is retained upon heating up to 900 °C [10].

In addition, the literature describes the preparation of anatase mesocrystals by kinetically controlled crystallization based on sol-gel synthesis. TiCl_4_ and n-octanol were used as reagents, and TiO_2_ mesocrystals in the form of a truncated bipyramid were obtained by oriented addition [10]. These particles have a large number of photoactive centers, and when irradiated with light in the visible region of the spectrum, they show high photoactivity, which exceeds the value for an industrial catalyst based on TiO_2_ (Degussa P25) [11].

In addition, TiO_2_ mesocrystals can be grown on substrates. For example, researchers [10] developed a method for synthesizing anatase mesocrystals on multiwalled carbon nanotubes (CNTs). CNTs were dispersed into an aqueous solution of TiF_4_, kept in an ultrasonic bath for 30 min, and then heated for 20 h at 60 °C. Nanotubes are coated with petal-shaped TiO_2_ particles, which consist of crystallites 2–4 nm in size. Metallic Ti can be used as a substrate, then rutile nanorods are synthesized in an aqueous solution of (C_4_H_9_O)_4_Ti and HCl at 150 °C for 20 h, while they are elongated along the [001] direction [10].

An analysis of the literature has shown that by now there are a significant number of experimental works devoted to the study of the photocatalytic properties of titanium dioxide, including the methods of its synthesis in the form of mesocrystals. Despite the amount of data available, interest in TiO_2_-based materials continues unabated, partly due to the search for affordable environmental cleanup materials. Furthermore, the thermal stability of anatase mesocrystals has not been yet reported.

In this study, preparation of photoactive oriented nanomaterials based on titanium dioxide and nonionic polymer was explored. Herein, we have presented a strategy to improve photocatalytic response of TiO_2_ composite.

The temperature limit of stability of an array of highly oriented anatase nanoparticles was confirmed, and the temperature of phase transition of titanium dioxide mesocrystals from anatase to rutile was determined for the first time.

## 2. Materials and Methods

### 2.1. Chemicals and Materials

(NH_4_)_2_TiF_6_, H_3_BO_3_ and PEG 400 were obtained from Sigma-Aldrich (Gillingham, UK). All chemicals were used without further purification.

### 2.2. Precursor (NH_4_TiOF_3_) Formation

First, NH_4_TiOF_3_ was synthesized via slow hydrolysis. The starting chemicals (NH_4_)_2_TiF6 (0.594 g) and H_3_BO_3_ (0.372 g) in the stoichiometric molar ratio 1:2 were mixed well with distilled water (30 mL) under stirring to form solution. After this, the PEG400 (9 g) solution was slowly added to prepared mixture under constant stirring. The viscous solution was kept at 35 °C for 20 h. The final products were collected by centrifugation at 7000 rpm for 7 min and then washed using H_2_O and acetone for three cycles. After drying at air sample NH_4_TiOF_3_ was obtained.

### 2.3. Mesocrystall and Layered Composites Formation

Obtained precursor (powder) were calcined in a muffle at certain temperatures (450, 600, 800, 1000 and 1200 °C) for 2 h at air atmosphere.

### 2.4. Analytics

The phase analysis was determined via X-ray diffraction (XRD) using a Bruker D8 Advance X-ray Diffractometer (Bruker, Germany). The morphology of the samples at all synthetic stages were recorded on scanning electron microscope Carl Zeiss NVision 40 (Carl Zeiss AG, Oberkochen, Germany) equipped with an INCA analyzer Oxford Instruments X-Max (Oxford Instruments, Abingdon, UK). Raman spectroscopy was conducted using a Renishaw inVia Reflex spectrometer (Renishaw, UK) with an illumination wavelength of 633 nm. Thermogravimetric and thermocalorimetric analysis was carried out using a Simultaneous Thermo Analysis Netzsch STA 409 PC Luxx (Netzsch-Gerätebau GmbH, Selb, Germany). Heating was conducted from room temperature to 1200 °C with a heating rate increase of 5 °C min^−1^ at air and argon. MS gas analysis was conducted at all temperature ranges. BET low-temperature nitrogen adsorption measurements were performed using an ATX-6 analyzer (Katakon, Novosibirsk, Russia). Measurements of photocatalytic activity were conducted under irradiation of a suspension of the analyte MC (1.5 mg MC in 2 mL H_2_O) in 40 µL aqueous solution of crystal violet (1 mM) in a quartz cuvette with an Ocean Optics HPX-2000 (Orlando, FL, USA) deuterium—halogen lamp (the output power is 1.52 mW, as measured in the 200–1100 nm range by an integrated optical power meter) in a cell thermostated at 37 °C. Spectrophotometric analysis was performed using an Ocean Optics QE65000 spectrometer (Orlando, FL, USA). All samples were kept in the dark for 45 min prior to conducting the degradation study. The chemical analysis was performed by X-ray photoelectron spectroscopy (XPS) using a SPECS spectrometer (SPECS Surface Nano Analysis GmbH, Berlin, Berlin) with Mg *K*α excitation (1253.6 eV).

Single-crystal data were collected at the ID28 diffraction side station of the European Synchrotron Radiation Facility, equipped with a PILATUS3 X 1M detector mounted on the rotating arm of a Euler goniostat [12]. In all cases the size of the focal spot was less than 50 mkm; wavelengths of 0.784 Å were employed in combination with different detector angles to ensure sufficient angular coverage. Single-crystal data were collected from isolated crystals of typical size ~5 × 5 × 1 µm^3^ mounted on glass fiber (Araldite Rapid epoxy). Dataset combined two subsets of 1440 images collected with 0.25 deg step in shutterless mode for two different angles of detector. CBF files were transformed to ESPERANTO format and treated by CrysAlis v38.41 software. High-resolution reciprocal space maps were produced by locally developed software of ID28 beamline. Images were binned by SNBL Tool Box software and integrated with the Dioptas program. Powder data refinement was carried out using FullProf.

## 3. Results

The structure, orientation and morphology of obtained NH_4_TiOF_3_ particles were investigated by XRD (including high resolution XRD) and SEM. Figure 1a shows the XRD diffraction pattern fully identical to [13]. The position and intensity of the reflections confirm the formation of single-phase crystalline NH_4_TiOF_3_. The shape of the obtained particles was established SEM: they are octagonal prisms with a base diameter of 3.8–5.6 µm and a thickness of 1.5–2.5 µm (Figure 1b). Figure 1c,d show the results of high-resolution X-ray diffraction of the precursor carried out on an individual macroparticle. The point diffraction pattern with intense ordered reflections confirms that the (001) axis of the crystallographic structure of NH_4_TiOF_3_ is coaxial with the normal to the base of the macroparticle prism.

In order to determine the optimal annealing temperature of the NH_4_TiOF_3_ precursor (PEG 400) to obtain anatase mesocrystals and study the processes of its transformations upon heating, thermal analysis with mass spectrometry was carried out (Figure 2).

Regarding the hypothesized reactions of the thermal decomposition of the precursor [8] and the obtained experimental data, an optimal temperature of 450 °C was chosen for the synthesis of anatase mesocrystals. To obtain TiO_2_ mesocrystals, the precursor obtained in the presence of PEG400 was annealed at 450 °C for 2 h in air. Figure 3a shows the X-ray pattern of the obtained substance, which confirms the complete conversion of NH_4_TiOF_3_ into crystalline titanium dioxide in the form of anatase without impurities. The Raman spectrum confirms the presence of the corresponding anatase phase (Figure 3b).

According to scanning electron microscopy (Figure 4) and X-ray diffraction for one single TiO_2_ microparticle (Appendix A, see Appendix A), it was found that all anatase nanoparticles are rectangular plates 30–50 nm in size, substantially co-oriented in the (001) plane. This is indicated by the high-resolution XRD patterns of the HK0 and the H0L reciprocal space layers close to the point one (Appendix A). The crystallographic axes of titanium dioxide (100) and (010) are aligned with the diagonals of the nanoparticle array. According to nitrogen adsorption data, the specific surface area for the obtained samples after evaluation via the BET model was 10.4 ± 5.0 m^2^/g.

It is known that the anatase–rutile phase transition temperature in a bulk crystal is close to 600 °C [14]. To determine the thermal stability of anatase in the form of mesocrystals, the NH_4_TiOF_3_ precursor synthesized using PEG 400 was annealed in air for 2 h at different temperatures: 600 °C, 800 °C, 1000 °C, and 1200 °C. According to the X-ray pattern (Figure 3a), anatase remained the only phase of TiO_2_ up to 1000 °C, while the X-ray diffraction pattern of sample annealed at 1200 °C contained only reflections of the rutile phase. Thus, the anatase–rutile phase transition in mesocrystals occurs in the temperature range 1000 °C–1200 °C, which is over 400 °C higher than transition of bulk TiO_2_ [14]. The increased thermal stability of the anatase phase can be explained by stabilization by fluorine anions, which remained in a small amount after heat treatment from NH_4_TiOF_3_. This effect is related to the high Ti–F binding energy, and, according to [15], the phase transition in anatase nanoparticles begins at a temperature of 900 °C. In our work, the mesocrystalline anatase phase is preserved in its pure form up to 1000 °C, and the phase transition occurs completely in the temperature range of 1000–1200 °C, which is higher than the available literature data [2]. The lattice constants for the sample annealed after 1200 °C (rutile TiO_2_) were evaluated as a = 4.5954(5) Å and c = 2.9599(8) Å, while the TiO_2_ derived from other material derived from precursor without fluorium demonstrate a = 4.591 Å and c = 2.957 Å [16]. The lattice parameters of TiO_2_ rutile phases were almost the same. It should be noted that the lattice parameters for mesocrystalline anatase differed from those for TiO_2_ obtained from precursor without fluorium: a = 3.791(1) Å vs. 3.785 Å and c = 9.48(0) Å vs. 9.514 Å [16]. It has been implicitly proved that fluorine can disappear after high-temperature anneal. The XPS analysis showed the presence of around 5–6% F in the nanoparticles of anatase obtained by annealing at 450 °C while there was an absence of F in the powder after 1200 °C heat treatment (Appendix A, see Appendix A). It was assumed that the F remaining in the structure would hinder the phase transition, but XPS failed to detect F for the samples annealed at temperatures of 1000 and 1200 °C. Probably, the amount of residual fluorine was below the threshold value for determining the composition of fluorine by this method (that is, below 1–3%). At the same time, the heat of the anatase–rutile phase transition was not high enough, and according to DSC data (Figure 2), it is difficult to determine the temperature range in which this transformation occurs for mesocrystals [2]. The increased stability of the anatase phase in highly oriented nanoparticle array can also be explained by the presence of residues of a rigid polymer matrix, which hinders the phase transition.

On both diffraction patterns and Raman spectra of the samples after annealing at temperatures of 800, 1000 and 1200 °C, an impurity was observed, identified as potassium hexatitanate K_2_Ti_6_O_13_ [17,18,19] (Figure 3b). The microstructure of the powders obtained after annealing at temperatures of 600, 800, 1000 and 1200 °C was studied by scanning electron microscopy (Figure 5). According to the micrographs, the shape of the aggregates was retained after annealing at 1000 °C. After heat treatment at 1200 °C, the geometry of the macroparticles was disturbed: the crystallites coarsened, which is typical for the transformation from anatase to rutile phase. The micrographs also showed a small amount of nanowhiskers, presumably of the K_2_Ti_6_O_13_ phase. A similar result had been obtained earlier in the synthesis of anatase mesocrystals in a PEG 6000 matrix after annealing at a temperature of 500 °C [20]. This component was formed as a result of potassium diffusion between the TiO_2_ layers, which was confirmed by the growth of nanowires along the (100) and (010) planes. The formation of the TiO_2_–K_2_Ti_6_O_13_ hybrid material with different morphologies was presumably due to the presence of potassium in the PEG 400 polymer matrix. TiO_2_ is a well-known ion-change nanomaterial [21]. The presence of potassium in the samples was confirmed by EDX (Appendix A, see Appendix A). K/Ti ratio was about 0.027.

All obtained samples were photoactive. A commercial TiO_2_ photocatalyst, Degussa P25, was used as a reference sample (Appendix A, see Appendix A and Table 1). A high photoactivity was established for a titanium dioxide nanomaterial obtained after annealing at 800 °C. The obtained indicators of photocatalytic activity (PCA) of the samples can be explained by the influence of two factors.

On the one hand, the appearance of the potassium hexatitanate (K_2_Ti_6_O_13_) phase significantly increased the photoactivity response. Thus, the PCA of mesocrystals after annealing at 800 °C was 4.4 times higher than after annealing at 450 °C. A number of works [15,18,22,23] present data on the high photocatalytic activity of K_2_Ti_6_O_13_. It was also shown that the photoactivity of the TiO_2_–K_2_Ti_4_O_9_ hybrid material was almost two times higher than that of pure TiO_2_ [24]. As for synthesis, the formation of K_2_Ti_6_O_13_ from TiO_2_ required temperatures above 800 °C or a long duration of synthesis [25,26,27,28], while the formation of the TiO_2_–K_2_Ti_6_O_13_ hybrid material in mesocrystals occurred upon heating for 8 h at a temperature of 500 °C [20]. On the other hand, according to micrographs (Figure 5), it was found that the highly oriented nanostructure of TiO_2_ began to destroy at 1000 °C, and after annealing at 1200 °C. The XRD and Raman spectroscopy data (Figure 3) make it possible to unambiguously conclude that there is a rutile phase in the powder, which has a lower photocatalyst [29].

## 4. Conclusions

In summary, preparation of photoactive oriented nanomaterials based on titanium dioxide and nonionic polymer was successfully realized. The study demonstrated the stability of obtained layered TiO_2_-anatase close to 1200 °C. This research for the first time verified that the transition of the mesocrystalline anatase to rutile phase occurs between 1000 °C and 1200 °C, which is over 400 °C higher than transition of bulk TiO_2_ due to the used precursors. A small amount of K-phase nanowhiskers, which was issued after 800 °C in the TiO_2_-based composite, stimulated the enhancement of the photocatalytic performance four times. This capability offers a new approach to constructing composite Ti-O layered materials and provides strong potential for the resulting nanomaterial for application in catalysis and electrode materials.

## Figures and Tables

**Figure 1 nanomaterials-12-04418-f001:**
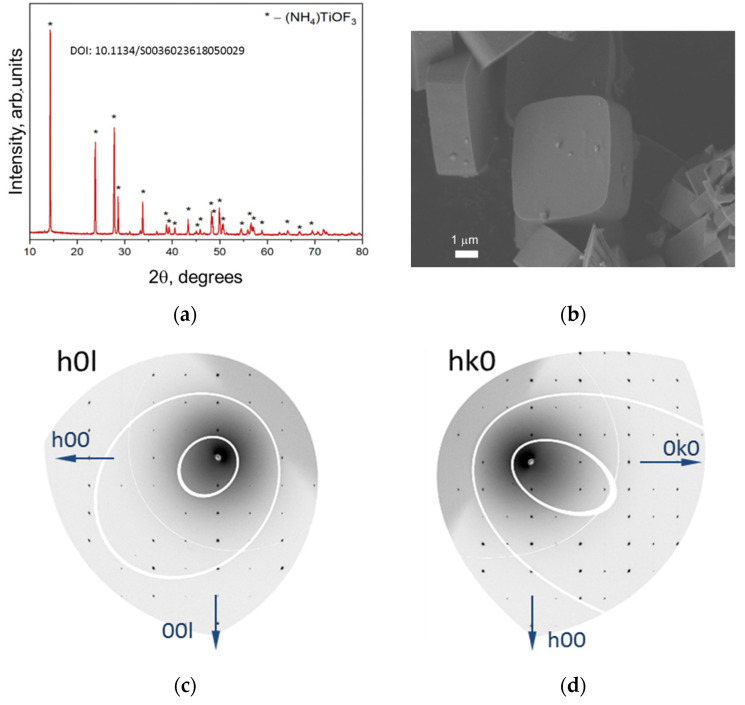
(**a**) Powder X-ray diffraction pattern of NH_4_TiOF_3_ precursor. (**b**) SEM micrographs of NH_4_TiOF_3_ mesocrystals formed in the presence of PEG-400. Scale bars are 1 µm. (**c**,**d**) images showing the HK0 and the H0L reciprocal space layers of an individual crystal of NH_4_TiOF_3_, respectively. Data were collected at the Swiss-Norwegian beamline BM01 at the European Synchrotron Radiation Facility (ESRF) in Grenoble, France.

**Figure 2 nanomaterials-12-04418-f002:**
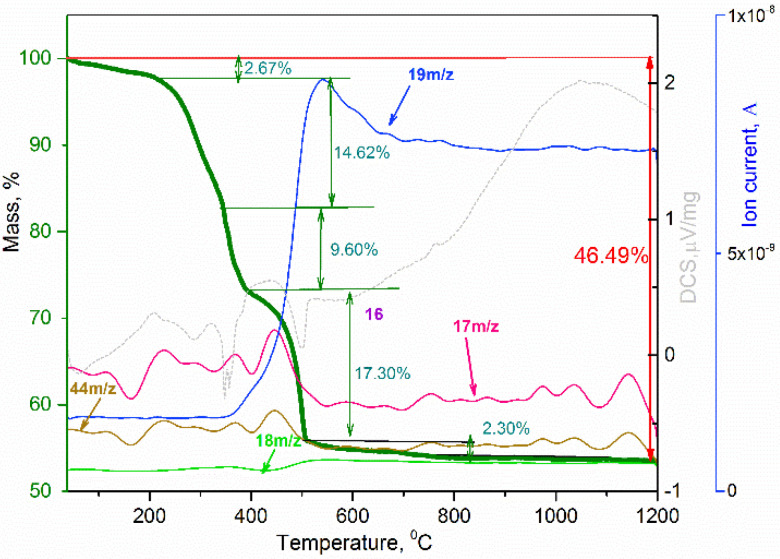
TGA-MS analysis of NH_4_TiOF_3_ at air atmosphere.

**Figure 3 nanomaterials-12-04418-f003:**
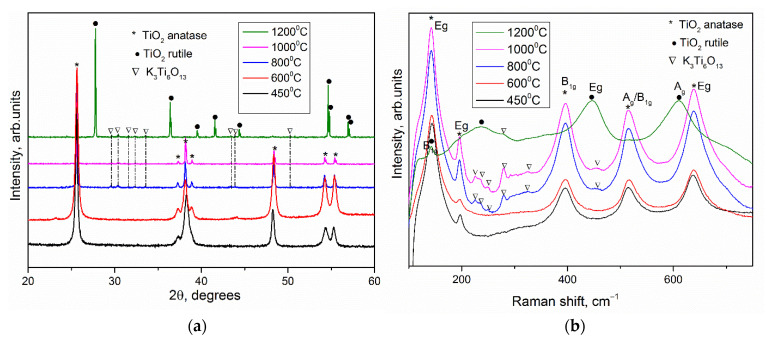
(**a**) XRD Patterns and (**b**) Raman spectra of TiO_2_ obtained by heat treatment of ammonium oxofluorotitanate after 450, 600, 800, 1000 and 1200 °C for 2 h. Phases identity as K_2_Ti_6_O_13_ (PDF #13-0447), anatase TiO_2_ (PDF #21-1272), TiO_2_ rutile (PDF #21-1276).

**Figure 4 nanomaterials-12-04418-f004:**
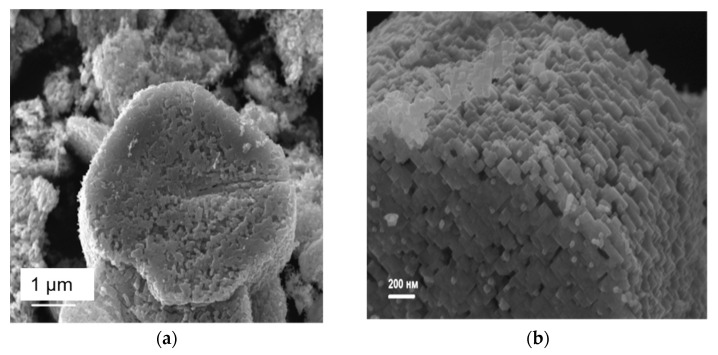
Scanning electron micrographs of a sample of TiO_2_ mesocrystals generated from a sample of NH_4_TiOF_3_ by heating under argon at 450 °C for 2 h: (**a**) showing porous layered structure of a collection of mesocrystals (white scale bar = 1 µm) and (**b**) an expanded image of a single TiO_2_ mesocrystal showing individual component nanoparticle building blocks comprising the mesocrystal (white scale bar = 200 nm).

**Figure 5 nanomaterials-12-04418-f005:**
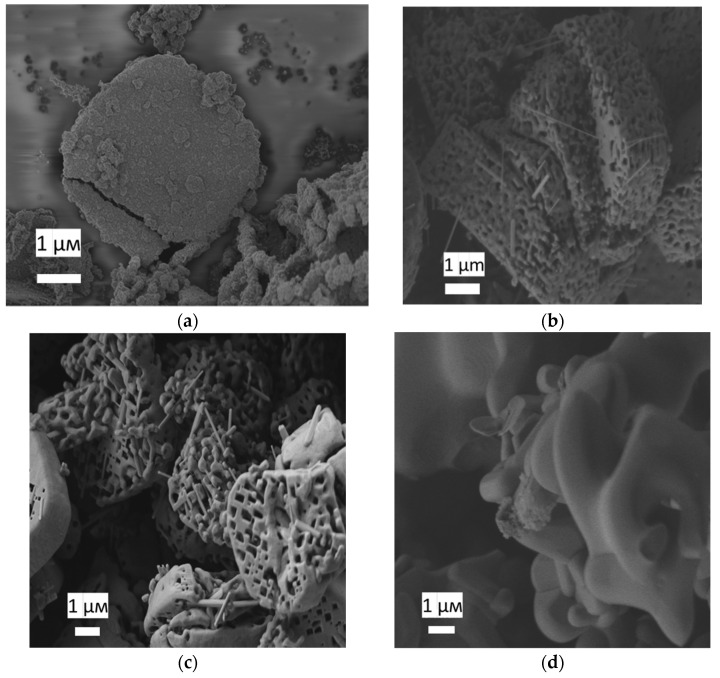
Scanning electron micrographs of a sample of TiO_2_ mesocrystals generated from a sample of NH_4_TiOF_3_ by heating under air (**a**) at 600 °C, (**b**) at 800 °C, (**c**) at 1000 °C and (**d**) at 1200 °C (white scale bar = 1 µm).

**Table 1 nanomaterials-12-04418-t001:** Phase content and photoactivity under UV of the obtained powders and commercial TiO_2_.

Scheme	Phase Content	Photocatalytic Activity
TiO_2_ after 450 °C	anatase	0.11%/min
TiO_2_ after 600 °C	anatase	0.21%/min
TiO_2_ after 800 °C	anatase + K-phase	0.48%/min
TiO_2_ after 1000 °C	anatase + K-phase	0.20%/min
TiO_2_ after 1200 °C	rutile + K-phase	0.02%/min
P25 (commercial)	anatase + rutile	0.63%/min

## Data Availability

Not applicable.

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
