# Peer review of "The Anatase-to-Rutile Phase Transition in Highly Oriented Nanoparticles Array of Titania with Photocatalytic Response Changes"

_nanomaterials, 2022, doi:10.3390/nano12244418_

Round 1

Reviewer 1 Report

The manuscript about phase tranisition of anatase towards rutile at about 1200C is for sure interesting as it usually occurs several hundreds C lower. The Authors are not delivering convincing mechanism of this stabilization, but anyway the presented material is worth of publication as it can pave the way for new synthesis methods of photocatalytical materials based on titania.

The several small issues should be addressed prior publication.

Line 76 - are elongated along the (001) direction – miller direction is usually denotedby using [001] vector.

97 distil water – should be „destilled water”

98 and 99 line (9g) 20h – should be “(9 g); 20 h” The space is required betwine the number and the unit. Please apply in the whole manuscript this rule

107 The morphology the samples – should be “The morphology of the samples”

128 - of typical size ~5x5x1 m3” the crystal seams a little to big to me ?. I guess the unit was “mm3”. Please also apply the superscript for cubic notation.

Figure 1a; Fig 3;  – the ordinate axis unit should be “arb. unit” as a.u. is reserved for astronomical units.

Figure 1b – please increase brightness or contrast. Now it is too dark.

Line 160 – X-ray pattern and Fig. 3a – please provide the number of PDF card ascribed to the identified phases. It will greatly help the reader interested in your XRD results.

Figure 4b and figure 5 – please increase the brightness. BTW – they look nice.

The Authors claim that the fluorium might hinder the phase transition to rutile phase. The samples annealed at temperatures 1000 and 1200C revealed absence of fluorium. However I suggest the determination of lattice constant for rutile TiO2 and compare it for TiO2 derived from other material derived from precoursor withouth fluorium. It is not direct proof, but might be good direction of proving your claim.

Supplementary materials:

Please improve the quality of EDS spectrum (Figure 2S) The ordinate axis description is missing tha abscissa delivers only unit of photon energy, but description is missing.

Reviewer 2 Report

The manuscript by Boytsova and co-Authors deals with the synthesis of anatase TiO2 using NH4TiOF3 as a precursor. Prepared materials were thermally treated at high temperature to investigate the stability of the anatase TiO2 polymorph. Photocatalytic activity tests were also assessed. 

The abstract and the Introduction sections of the manuscript are very difficult to read. They should be re-written to be more clear to the reader. 

HR-TEM investigations are missing as an integration to XRD to prove an actual anatase oriented nanoparticles (along which crystallograpic direction?). 

Photocatalytic tests shall be describe with more detail. Which was the initial dye concentration? Which the shape/internal volume of the reactor used? At which wavelength did the Authors assessed the measurements in the spectrometer? Which is the contribution of that potassium titanate in the photocatalytic activity? BET measurements are missing too. 

Minor:

Abstract Line 20: What do Authors mean with mesocrystalline here?

Abstract Lines 21-22: "which is over 400°C higher than transition of bulk TiO2 due to the initial." Initial what? Something is missing here. 

Figure 3a: Authors should include the pattern in the caption follow a certin temperature order. Not randomly. As it is, the legend is confusing.

Table 1, last column: dot to separate the decimal figures. 

Authors talks about XPS (page 6 line 192), but there is no mention of it in the Experimental. 

For the above reasons, the paper is not suitable for publication in Nanomaterials
